# Ultra-Low-Power Digital Filtering for Insulated EMG Sensing

**DOI:** 10.3390/s19040959

**Published:** 2019-02-24

**Authors:** Theresa Roland, Sebastian Amsuess, Michael F. Russold, Werner Baumgartner

**Affiliations:** 1Institute of Biomedical Mechatronics, Johannes Kepler University Linz, 4040 Linz, Austria; werner.baumgartner@jku.at; 2Research and Development, Otto Bock Healthcare Products GmbH, 1110 Vienna, Austria; sebastian.amsuess@ottobock.com (S.A.); michael.russold@ottobock.com (M.F.R.)

**Keywords:** EMG signal processing, biosignal processing, insulated/capacitive EMG, low power filtering, myoelectric upper-limb prosthesis

## Abstract

Myoelectric prostheses help amputees to regain independence and a higher quality of life. These prostheses are controlled by state-of-the-art electromyography sensors, which use a conductive connection to the skin and are therefore sensitive to sweat. They are applied with some pressure to ensure a conductive connection, which may result in pressure marks and can be problematic for patients with circulatory disorders, who constitute a major group of amputees. Here, we present ultra-low-power digital signal processing algorithms for an insulated EMG sensor which couples the EMG signal capacitively. These sensors require neither conductive connection to the skin nor electrolytic paste or skin preparation. Capacitive sensors allow straightforward application. However, they make a sophisticated signal amplification and noise suppression necessary. A low-cost sensor has been developed for real-time myoelectric prostheses control. The major hurdles in measuring the EMG are movement artifacts and external noise. We designed various digital filters to attenuate this noise. Optimal system setup and filter parameters for the trade-off between attenuation of this noise and sufficient EMG signal power for high signal quality were investigated. Additionally, an algorithm for movement artifact suppression, enabling robust application in real-world environments, is presented. The algorithms, which require minimal calculation resources and memory, are implemented on an ultra-low-power microcontroller.

## 1. Introduction

Electromyography (EMG) is the measurement of electrical potential arising from electrochemical effects due to muscle contractions. These signals are transmitted via human tissue to the surface of the skin, where they can be measured by surface EMG electrodes. Areas of application for these sensors include exoskeletons, diagnostics and myoelectric hand prostheses. The state-of-the-art electrodes in prostheses use a conductive connection to the skin [1,2,3]. Numerous EMG systems for myoelectric hand prostheses with various levels of dexterity have been previously presented [4,5,6]. However, in practice, many amputees reject such high-dexterity myoelectric prostheses because they require high cognitive effort [7]. Many patients prefer a basic hand prosthesis which is robust and easy to handle [8]. The algorithms which we present in this article are designed for a conventional myoelectric hand prosthesis with two EMG sensors and two degrees of freedom as provided by Otto Bock Healthcare Products GmbH [2].

In contrast to a conductive electrode, the insulated EMG sensor couples the muscle contraction signal from the skin to the sensor electronics capacitively. The signal quality in conductive sensing depends on the presence of an electrolyte in between the skin and the sensor electrode, which consists of sweat (dry) or an electrolytic paste (wet). The impedance of the stratum corneum decreases with time [9,10] at dry electrodes and therefore requires gain level adjustments when exploiting the full operating range. The conductive sensors often have protruding metal parts, which can cause pressure marks. These are problematic, especially for patients with circulatory disorders. For these reasons, we suggest insulated EMG sensors. They have high signal quality immediately after applying the sensor. No gain level adjustments are necessary due to the sweat independence. Pressure marks are avoided as flexible sensors were designed; textile sensors are especially comfortable to the skin. Compared to the conductive measurement principle the capacitive measurement principle has different requirements and higher effort for the sensor electronics and the signal processing. The electronics for this highly stable, low-power EMG sensor for real-world applications have already been published by Roland et al. [11]. Here, we present the associated digital signal processing by an ultra-low-power microcontroller (μC) embedded in the sensor electronics. The digital signal processing was developed for a low-level dexterity system with only two electrode sites, which is common in most of the current commercial upper-limb prostheses [2,12]. However, these algorithms can be adjusted to suit other applications too.

Numerous publications have suggested optimal cutoff frequencies for EMG bandpass filtering for conductive EMG sensors. The significant power spectrum of EMG signals ranges from approximately 20–500 Hz [13,14]. Low-frequency noise, such as movement artifacts, occurs predominantly in the range of 0–20 Hz [15,16]. The recommended cutoff frequency *f_C_* for the highpass filter for attenuating these low-frequency artifacts is within the range of 5–30 Hz for conductive EMG sensors [16,17,18]. A 400–500 Hz lowpass filter *f_C_* is recommended for filtering high-frequency noise while maintaining EMG signal power [16,17,18].

Digital filtering realized by programmable μC has various advantages over analog filtering [19]. Active analog filters of second or higher order require many components [20], which must be matched exactly. Component accuracy and the resulting filter behavior is limited by manufacturing, temperature and aging tolerances. Providing various analog filters requires several active and passive low-noise high-precision electronic components, which increase power consumption and cost. One might argue that analog filters are faster; however, in the case of EMG at a sampling frequency in the kHz-range, the delay caused by the presented digital signal processing is negligible. The signal delay that is dominated by the time constant *T* of the lowpass filters does not depend on the realization of the filter. Anyway, an analog anti-aliasing filter upstream the analog-to-digital conversion (ADC) to conform with the Nyquist–Shannon sampling theorem [21] is indispensable. The digital filter algorithms can simply be adjusted. For example, the notch frequency for the power-line interference can be changed by one parameter from 50 Hz in Europe to 60 Hz in North America. Furthermore, it allows the real-time implementation of additional algorithms like artifact suppression or dexterous prostheses control. Such adjustments require re-engineering and a new design of the sensor electronics at analog signal processing.

In this article, we propose parameters for highpass, comb and lowpass filters for a low-cost, compact-size and insulated EMG sensor. Moreover, we present high-speed algorithms for rectification and smoothing of the EMG signal for myoelectric prosthesis control. Movement artifacts and other noise are considered in the evaluation of the optimal filter parameters to obtain a robust system for real-world environments. The filters were implemented on an ultra-low-power μC for real-time signal processing at minimal computational cost. Since the μC does not contain a floating-point unit, the filter coefficients and signal processing were implemented in fixed-point representation. Hence, fast calculation by the single-cycle 32 bit hardware multiplier integrated in the μC can be exploited. Quantization and overflows due to range limits were considered in the filter design.

The digital signal processing should meet the following requirements:

**High Signal Quality:** A high signal-to-noise ratio (SNR) should be achieved.

**Stability:** The prosthesis drive should not be activated by movement artifacts or other noise.

**Efficient Calculation:** Digital signal processing must be implemented for real-time operation such that it minimizes CPU calculation resources.

**Short Response Time:** Farrell et al. [22] stated that controller delays of more than 100 ms lead to a decrease in the performance of prosthesis control. The delay introduced by the sensor should be as short as possible, but certainly less than 100 ms.

**Optimized Memory Requirements:** Low memory consumption allows the implementation of additional algorithms that classify hand movements [4,6] or discriminate between contraction EMG and artifacts [23].

**Proportionality:** In a conventional myoelectric prosthesis, the gripping speed, respectively the gripping force, is controlled by the differential signal of two EMG sensors, which is considered to be proportional to the muscle force. Therefore, the EMG signal must be processed such that the sensor output signal is proportional to the strength of the muscle contraction.

**Co-contraction:** When two EMG sensors control the prosthesis, a simultaneous short and strong contraction of both muscle groups is called co-contraction. The slopes of the rectified and smoothed signal must be sufficiently steep and the peak amplitude sufficiently high to enable co-contraction. Co-contraction toggles between different prosthesis functions. In our specific case, it toggles the prosthesis movement control between open-close and supinate-pronate. 

In this article, we first explain the problems at insulated EMG measurement. Second, we describe the measurement set-up for EMG signal acquisition, followed by the digital signal processing including comb, lowpass and highpass filtering. Furthermore, we present the filter architectures and their implementation in C. The digital signal processing in our sensor also includes rectification, smoothing and a decision algorithm for distinguishing between contraction EMG signal and artifacts. The results section presents the optimal parameters for the filters as well as their stability and runtime.

## 2. Problem Definition

The choice of filters for noise that overlap with the EMG frequency range is crucial. The following noise occurs in biosignal measurements [15]:

### 2.1. Noise Caused by Electronic Components

Noise caused by electronic components affects the whole frequency range. In the EMG frequency range, it can only be reduced by selecting low-noise components and circuits in the measurement set-up.

### 2.2. High-Frequency Noise

Noise from high-frequency electromagnetic radiation due to mobile phones, WLAN, TV and electronic components are filtered by an analog lowpass with a cutoff frequency of 1064 Hz [11]. When the analog signal is sampled digitally, this lowpass is indispensable in preventing aliasing.

### 2.3. Power-Line Interference (PLI)

The power-line frequency with 50 or 60 Hz and its harmonics overlap with the EMG frequency range (Figure 1). Their amplitudes can exceed the EMG signal significantly. In this contribution, we filter the power-line frequency at 50 Hz and its harmonics digitally. In addition, 60 Hz comb filters can be designed in the same way.

According to the EN 50160 standard [24], the power-line frequency is supposed to deviate within the range of 49.5–50.5 Hz for 99.5% of the time in Europe. For a robust system, these deviations must be considered when selecting an appropriate comb filter. In this work, we evaluated various filter designs for PLI suppression while maintaining maximum EMG signal power.

### 2.4. Movement Artifacts

In biosignal measurement, low-frequency artifacts (i.e., movement artifacts) are predominantly in the range of 0–20 Hz [15,16]. Relative movements of the muscle to the EMG sensor generate these artifacts, for example, at the beginning and at the end of a contraction. In capacitive EMG sensors, changes in contact pressure change the capacitance between the signal source and the sensing electrode (i.e., the coupling capacitance) [11]. This leads to asymmetries in differential EMG sensor measurement and thus to movement artifacts, which can also result from movements of the cable connecting the electrode to the amplifier. Figure 2a shows a typical artifact in the time domain, measured with the EMG sensor described in Roland et al. [11] and Figure 2b shows a qualitative sketch in the frequency domain.

The flexible design of the sensor assemblies reduces movement artifacts because they adapt to the human forearm anatomy. The capacitive measurement principle already leads to a highpass characteristic [11]. Additionally, we implemented a first-order highpass with a cutoff frequency of 11 Hz in the analog circuit. In this article, we evaluate the choice of the cutoff frequency for the digital highpass filter for capacitively coupled EMG. A good balance between maintaining the EMG signal power and damping the movement artifacts must be found.

## 3. Measurement Set-Up

The digital signal processing was designed for the insulated EMG measurement set-up described in Roland et al. [11] (Figure 3), which was applied for EMG signal acquisition in this work. Figure 4 shows the application of the measurement set-up to the human forearm, the EMG sensor and the printed circuit board.

### 3.1. Interface to Human Body

Due to the insulating layer, the flexible sensing electrode has no conductive connection to the skin. This sensor is a multilayer construct made of textiles, foils or a flex circuit board. In this work, the measurements were performed with the foil sensor. The sensor area is shielded with a common-mode shield to increase the common-mode rejection ratio and to protect against noise from the environment. A small conductive reference made of textile connects the sensor electronics to the electrical potential of the body. This reference keeps the common-mode potential of the sensor input amplifier within operating range.

For the measurements presented in this article, the EMG sensor was placed at the musculus extensor digitorum, at one third of the distance between the epicondylus lateralis and the ulna. This muscle was selected for the experiments as a large proportion of upper-limb amputations are at trans-radial level or are farther distal [25,26,27,28].

### 3.2. Analog Circuit

The analog signal, which is coupled via the EMG sensing electrode, is amplified by the instrumentation amplifier by a gain of 26. The bandpass is formed by a first-order analog highpass and lowpass filter. The highpass has a cutoff frequency of 11 Hz to attenuate the low-frequency noise, and the lowpass has a cutoff frequency of 1064 Hz to filter the high-frequency noise. The signal is further amplified by the μC’s internal operational amplifiers (OpAmps) by software-programmable gain. This gain factor is varied to exploit the full operating range at the ADC input. For this work, the gain of the internal OpAmps was varied in the range of 4 to 128, depending on the experiment (noise for comb measurements 128, EMG signal and noise for signal to noise ratio measurements 128, EMG for highpass measurements 32, EMG for comb measurements 16, artifacts for highpass measurements 4) [11].

### 3.3. Analog-to-Digital Conversion (ADC)

The ADC samples and converts analog signals to digital values. The embedded ADC is configured to extend the default 12-bit resolution by accumulation and averaging to 16 bits. When accumulating 64 samples, the result is right-shifted automatically by two bits to fit the 16-bit register size. To this end, the resolution was increased from 12- to 16-bit precision and noise performance was improved by averaging four samples. In order to be deterministic and synchronous with the digital signal processing (DSP), the ADC result is read and a new ADC conversion is started by the Timer/Counter TC0 callback routine as described in Section 3.4.1.

### 3.4. Digital Signal Processing (DSP)

For DSP, we employed the ATSAML21E18B [29] from Microchip Technology Inc. (Chandler, AZ, USA) This is an ultra-low-power μC with a 32 bit ARM^®^ Cortex^®^-M0+ processor with a maximum clock frequency of 48MHz, 256 kB flash and 32 kB SRAM main memory. For our insulated EMG sensor, the 32 pin version was selected for easy prototyping of the circuit board. In the test set-up, the clock frequency was set in the range of 2MHz–48MHz. The controller has a 32 bit hardware multiplier, but no hardware divider and no floating-point unit on board. Figure 5 shows a block diagram of the DSP path. A previous version of the digital EMG signal processing with an ATSAML21 was published by Roland et al. [30].

#### 3.4.1. Cyclic Program Processing

Deterministic and cyclic program processing for the DSP functionality is realized by a timer interrupt. The onboard Timer/Counter TC0 is configured to invoke a callback routine, which executes the DSP algorithm. This cycle defines the DSP sampling rate $f_{S}$.

Larivière et al. [31] and Li et al. [32] state that a sampling frequency of 400–500 Hz is sufficient for EMG measurement. In contrast, Enderle [33] suggests to sample the measurement signal at frequencies at at least ten times the highest signal frequency. High signal quality was desired to evaluate the optimal filter parameters, therefore, the EMG sampling rate $f_{S}$ was chosen at 10 kHz in this work. A high sampling frequency avoids the reflection of high frequency noise, which is not sufficiently attenuated by the anti-aliasing filter, to the EMG frequency range. The proposed filter architectures and cutoff frequencies can correspondingly be applied at lower sampling rates, which allows a reduction of the clock frequency. In Section 4.8, we show the effects of reducing the sampling frequency.

#### 3.4.2. Fixed-Point Representation

*q15_t*, *q31_t* and *q63_t* are standard data types for fixed-point representation in C. Fixed-points, as described by ARM Ltd. (Cambridge, UK) [34], are implemented for the DSP. They resemble an integer data type, and their bit pattern must be used correctly. The coefficients are multiplied by quantization factors for correct scaling of the fixed-point values. Calculations with standard Q1.15 variables result in fixed-points with deviating bit pattern; for instance, a Q2.15 is generated when two Q1.15 variables are added. Therefore, the programmer must implement the data type with the correct bit pattern while considering truncation and overflow or saturation.

The numerical values must be scaled such that the full operating range is exploited while overflows are prevented. When calculating with fixed-point data type, the μC processor uses integer numbers, and thus the high performance of the integrated single-cycle hardware multiplier of the 32 bit ARM processor can be exploited. Quantization and overflows are taken into account in the filter design. To ensure stability after quantization, the poles of the quantized coefficients were investigated.

#### 3.4.3. Comb Filter

The comb filter was designed as a highpass filter with a sampling frequency of 50 Hz, which is the frequency to be filtered *f_filt_*. The amplitude spectrum is mirrored at *f_filt_*/2 according to the Nyquist–Shannon sampling theorem [21] and it is identical at its harmonics, which leads to a comb filter behavior. The name of the comb filter (1–7 Hz) is derived from the highpass cutoff frequency *f_C_* used to design the filter. Accordingly, the comb filter cutoff frequencies around 50 Hz are in the range of 43–49 Hz and 51–57 Hz. This frequency characteristic is replicated at the harmonics of 50 Hz.

In this work, we considered applying Bessel [35], Butterworth [36], Chebyshev [37] and Elliptic [38] filters. Butterworth and Chebyshev filters were implemented, as they provide a good balance between low ringing and overshoot and a steep roll-off. The Bessel and the Elliptic filters were not implemented because the former has low roll-off and the latter tends to ring and overshoot.

The filter was designed with the Matlab^®^ FDA filter tool [39] by first determining the filter zeros, poles and gain for an analog lowpass. These were then transformed into state-space form, where the lowpass can be converted to the desired filter behavior, such as highpass, bandpass or bandstop. By bilinear transformation [40], the analog filter was transformed into a digital filter and then converted back to zero-pole gain form. The filter coefficients determined were floating-point numbers (Table 1). Additionally, a third-order finite impulse response (FIR) filter which enables fast calculation was implemented for comparison. To this end, the frequency-independent scaling due to the filters was compensated for equal passband amplification. However, in the final implementation, this scaling does not need to be compensated at this point as long as the signal remains within value range of the fixed-point variable.

The filter frequency $f_{filt}$ = 50 Hz corresponds to a 20 ms time period, which equals 200 samples at the EMG signal sampling rate $f_{S}$ = 10 kHz. Figure 6 shows the transfer function
(1)$$G_{COMB}\left[z\right]=\frac{COMB_{out}\left[z\right]}{ADC_{signal}\left[z\right]}=\frac{b\left[0\right]+b\left[200\right]z^{-200}+b\left[400\right]z^{-400}}{a\left[0\right]+a\left[200\right]z^{-200}+a\left[400\right]z^{-400}}\;,$$
which considers the quantization effects.

#### 3.4.4. Comb Filter Evaluation

For the comb evaluation, various comb filters were implemented on the μC and assessed in a standardized scenario. Two test subjects were measured three times each with contracted and relaxed muscle. The measurements were repeated for all comb implementations. The amplitude at the DAC, to which the highpass output was applied, was measured with an oscilloscope at a 10 kHz sampling rate. These measurements were examined for the ability to attenuate PLI while maintaining EMG signal power.

#### 3.4.5. Comb Filter Implementation

The comb filter was implemented in C as an infinite impulse response (IIR) filter in direct form II (Figure 7), for which only one center array was required. The center array, with length =200filterorder+1, is a ring buffer [41], so the entries do not need to be shifted at each iteration. For addressing the ring buffer, a pointer array with a length of filterorder+1 is implemented.

This IIR filter has coefficients in a Q2.10 format, and they are implemented as a 16-bit fixed-point in Q6.10 format. The input has Q1.15 format. The coefficients and the input signal are multiplied. This intermediate result is stored in a 64-bit internal accumulator in Q39.25 format. Additions can therefore be executed without the risk of overflows while preserving full precision. The resulting center in Q5.25 format is then right-shifted by 10 bits, thus removing the least significant bits, and stored in a 32 bit fixed-point in Q17.15 format. The center for the second-order filters is calculated by
(2)$$\begin{array}{cc} center[k] & =(a\left[0\right]ADC_{signal}-a\left[200\right]center\left[k-200\right]-a\left[400\right]center\left[k-400\right])\gg 10.\; \end{array}$$

In the next step, the output is calculated from the 32-bit (Q17.15 format) center array by multiplications with the Q2.10 format coefficients. Again, a 64 bit internal accumulator eliminates the risk of overflows and preserves full precision. When these Q7.25 fixed-points are added up three times, a Q9.25 format is generated. This intermediate result is truncated again by removing the 10 least significant bits and generating a Q9.15 format. This is saturated to a Q1.15, which does not affect the contraction EMG, as it does not exceed the value range of the fixed-point variable; however, artifacts might exceed the value range. The Q1.15 output, which is stored as a 16 bit fixed-point variable, is calculated by
(3)$$\begin{array}{cc} 1-1COMB_{out} & =(b[0]center[k]+b[200]center[k-200]+b[400]center[k-400])\gg 10\;. \end{array}$$

##### Comb Filter Runtime

The runtime of the comb filter was measured with an oscilloscope [42]. For this purpose, only the comb filter function was implemented; its start and end are indicated by a digital output signal change.

#### 3.4.6. Highpass Filter

The highpass filter attenuates low-frequency artifacts, which are described in Section 2.4.

We evaluated which filter characteristics are appropriate for insulated EMG sensors. To this end, we designed second-order Chebyshev and Butterworth IIR filters. The sampling frequency *f_S_* was set to 10 kHz and the cutoff frequency *f_C_* varied from 20 Hz to 100 Hz with a 10 Hz iteration, which resulted in the coefficients listed in Table 2.

The numerators for all highpass filters are:(4)$$\begin{array}{c} b=\left[\begin{array}{ccc} 1 & -2 & 1 \end{array}\right]. \end{array}$$

As these coefficients are implemented in fixed-point format, the denominators and numerators are scaled and truncated. Figure 8 shows the transfer function for the highpass with the quantized coefficients:(5)$$G_{HP}\left[z\right]=\frac{HP_{out}\left[z\right]}{HP_{in}\left[z\right]}=\frac{b\left[0\right]+b\left[1\right]z^{-1}+b\left[2\right]z^{-2}}{a\left[0\right]+a\left[1\right]z^{-1}+a\left[2\right]z^{-2}}\;.$$

##### Highpass Filter Evaluation

To determine the optimal choice of filter characteristic and *f_C_*, we recorded EMG signals from three subjects both at the left and at the right arm. A 50 Hz comb filter was implemented at the μC for the measurements, but no highpass filter was implemented for the evaluation. Three contraction and three artifact signals were recorded at each arm of each subject. Each measurement lasted 10 s, but only 8 s were evaluated because the on-set of the contraction was not considered. The artifacts were created by mechanical interferences, which might occur in real-world environments, such as tapping, shifting or lifting of the sensor. By using these artifacts for the filter parameter evaluation, the resulting system will be robust against this noise. The entire 10 s of the artifacts were evaluated. The oscilloscope [42] was set to a sampling rate of 10 kHz.

The evaluation was performed in Matlab^®^ [39]. To this end, the filtering was carried out such that the μC implementation was represented. The coefficients, the measurement values, the intermediate results and the final results were quantized and truncated as in the controller. Furthermore, the limits of the variable sizes were considered in the filtering to prevent overflows.

Based on the root mean square value, we calculated the signal loss in % for all the contraction EMG and the artifact measurements for the different filters. The differences between artifact- and EMG signal losses were calculated, as the maximum value indicates an optimal trade-off.

The EMG signal was evaluated in the frequency range from 40 to 600 Hz to consider contraction EMG power, but no low-frequency noise. The choice of the 600 Hz had a negligible impact on the results as long as it exceeded the main EMG power frequency range.

##### Highpass Filter Implementation

The highpass with the maximum difference between artifact signal loss and EMG signal loss was implemented in direct form II (Figure 7) at the μC. The bit pattern of the fixed-point data type of the input, coefficients, intermediate results and output is equivalent to the comb filter implementation in Section 3.4.5. The implementation was done according to
(6)$$\begin{array}{cc} center[k] & =(a\left[0\right]HP_{IN}\left[k\right]-a\left[1\right]center\left[k-1\right]-a\left[2\right]center\left[k-2\right])\gg 10\;, \end{array}$$
(7)$$\begin{array}{cc} HP_{out}\left[k\right] & =(b[0]center[k]+b[1]center[k-1]+b[2]center[k-2])\gg 10\;. \end{array}$$

##### Highpass Filter Runtime

The highpass filter runtime was calculated in the same way as the comb filter runtime (see Section “Comb Filter Runtime”).

#### 3.4.7. Lowpass Filter

Two lowpass filters are applied in the DSP. One eliminates the high frequency noise after the highpass, and the other one is applied for smoothing to get a DC output signal, which is proportional to the muscle contraction intensity to control the prosthesis drive.

We designed a first-order IIR filter (first-order lag element) with the transfer function:(8)$$G_{LP}\left[z\right]=\frac{Y[z]}{U[z]}=\frac{b[0]}{a[0]+a[1]z-1}$$
with the input U[z] and the output Y[z].

This lowpass filter was designed such that it requires only one storage variable. Therefore, the filter coefficients are described with the parameter *c* as follows:(9)a=[1,−c],
(10)b=[c].

The coefficients *a*[1] and *b*[0] are selected with equivalent absolute values to facilitate the calculation, as only one multiplication is required:(11)y[k]=(u[k]+y[k−1])c
with the preliminary output y[k], the input u[k] and *c* as the coefficient.

The behavior of first-order lag elements is adapted by the time constant *T* or the cutoff frequency $f_{C}$, which relate to the parameter *c* at this lowpass as follows:(12)$$T=\frac{c\Delta t}{1-c}=\frac{1}{2\pi f_{C}}\;,$$
(13)$$f_{C}=\frac{1-c}{c2\pi \Delta t}\;.$$

Hence, the coefficient *c* is calculated according to:(14)$$c=\frac{1}{1+\frac{\Delta t}{T}}=\frac{1}{1+2\pi f_{C}\Delta t}$$
with the sampling time
(15)$$\Delta t=\frac{1}{f_{S}}\;.$$

The filter introduces a scaling
(16)$$S=c(1+\frac{T}{\Delta t})\;,$$
which is compensated for by a shift of the preliminary output before being passed to the next signal processing stage:(17)$$LP_{OUT}\left[k\right]=y\left[k\right]\gg round\left(log_{2}\left(S\right)\right).\;$$

Figure 9 shows the resulting transfer function in the frequency domain *g_LP_*[*f*] and the transfer function including a shift for scaling. Note that compensation for the scaling does not necessarily have to be implemented directly after the lowpass filter. A required shift can be combined with other multiplications or shift operations.

##### Lowpass Filter Implementation

For the μC implementation, the preliminary output y[k] was calculated according to Equation (Equation 11). Only one addition, multiplication and shift are required at each sample.

The preliminary output y[k] was implemented as a 64-bit fixed-point variable in C. The input u[k] (Q1.15) and the coefficient *c* (Q1.10) were implemented as 16-bit fixed-point variables. The result is right-shifted by 10 bits at each sample step, thus removing the 10 least significant bits. The lowpass output $LP_{OUT}\left[k\right]$ is implemented as a Q1.15 variable.

In the lowpass filter after the highpass, no saturation was implemented for y[k] and $LP_{OUT}\left[k\right]$, since the EMG signal is distributed across the operating range (no clipping) and therefore does not cause overflows. For artifacts, which may cause clipping at the limits of the value range, overflows may occur. As the prosthesis drive is turned off for artifacts in any case, these overflows are tolerated.

For filtering the signal after the highpass filter, *f_C_* was set to 531 Hz, which results in a *c* of 0.75. y[k] was right-shifted by one bit to calculate the output $LP_{OUT}\left[k\right]$. In the lowpass for smoothing, the *f_C_* was set to 3.1 Hz (*c* = 0.9981) and the signal was right-shifted by eight bits.

##### Lowpass Filter Runtime

The measurement of the runtime is described in Section “Comb Filter Runtime”. The runtime of (i) the described lowpass filter with fixed-point data type was compared to (ii) an implementation with fixed-point data type in direct form II (Figure 7) and (iii) an implementation with floating-point data type in direct form II.

(i) A first-order IIR filter with fixed-point data type as designed in Section 3.4.7.

(ii) A first-order IIR filter was implemented with fixed-point coefficients in direct form II (Figure 7) for a cutoff frequency *f_C_* of 3.1 Hz. In contrast to the presented architecture, this implementation requires more arithmetic steps. The center is a 32-bit array and the coefficients, input and output are 16-bit fixed-point variables.

(iii) For the floating-point implementation, a first-order IIR filter was designed and implemented in C. The *f_C_* was set to 3.1 Hz, which is equivalent to the smoothing *f_C_*. The input, the coefficients, the center array of the direct form II implementation and the output are 32-bit floating-point numbers (*float32_t*). The value range of the variables was chosen such that overflows are prevented.

(iv) Additionally, a FIR filter with floating-point coefficients (*float32_t*) was designed and implemented in C. The input, center array and output are also in 32-bit floating-point format. The filter wase designed with order 5, and the *f_C_* was set to 3.1 Hz.

(v) Same filter as (iv) with filter order 8.

The floating-point filters are not described in more detail, as they were implemented only for the runtime comparison.

#### 3.4.8. Rectification and Smoothing

The EMG signal is rectified and smoothed to control the prosthesis drive.

Linear rectification
(18)$$rect_{out,linear}\left[k\right]=abs\left(rect_{in}\left[k\right]\right)$$
and squared rectification
(19)$$rect_{out,squared}\left[k\right]=rect_{in}{\left[k\right]}^{2}$$
were considered in this work. A combination of these two would be possible to combine their behavior, but, due to its higher calculation effort, it was not implemented.

After rectification, the signal is limited to prevent overflows and keep the signal within a defined operating range. Due to this limit, the signal returns faster to amplitudes below the activation threshold after a contraction.

Smoothing is achieved with the lowpass filter ($f_{C}$ = 3.1 Hz) as described in Section 3.4.7.

After smoothing, the signal is right-shifted by eight bits. An offset of 113 is added, which corresponds to 91 mV. The output signal to the prosthesis drive will be proportional to the muscle contraction as described in the Introduction.

##### Rectification and Smoothing Runtime

For the runtime calculation, see Section “Comb Filter Runtime”.

#### 3.4.9. Decision Algorithm

The decision algorithm is not considered in detail in this article, as it has already been described by Roland et al. [23].

Prostheses drive activation by movement artifacts that have not been filtered sufficiently by the highpass is one of the major problems in myoelectric prosthesis control. This has to be avoided to the greatest feasible extent. For this reason, a real-time decision algorithm was implemented. The short-time Fourier transform (STFT) of signal windows is calculated in the μC; these windows overlap in time for fast decisions by the algorithm. A compromise between good frequency and good time resolution must be found. Narrow STFT windows lead to a good time resolution but poor frequency resolution, and vice versa. The algorithm decides whether the signal is an artifact or a contraction, which results in disabling or activation of the prosthesis drive. The amplitude spectrum of a reference contraction is determined by an automatic calibration procedure and stored in the non-volatile memory on the μC. Frequency bands of the current STFT signal are compared with the reference amplitude spectrum. The difference between reference and current signal is calculated for each frequency band, and then the sum of these differences over the frequency bands is computed (gray area in Figure 10). When this sum exceeds a predefined threshold, the prosthesis drive is disabled to avoid erroneous activation.

### 3.5. Output

For evaluation of the measurement set-up, the digitally filtered signal can be measured at the DAC with an oscilloscope [42], or the digital signal can be transmitted to a PC via Bluetooth Low Energy (BLE). In real-world applications, the filtered, rectified and smoothed signal is passed on to the prosthesis drive via the 12-bit digital-to-analog converter (DAC). The DAC is configured to use the analog supply voltage as internal reference voltage resulting in a conversion range between ground (0 V) and analog supply voltage (3.3 V). A new conversion starts as soon as a new value is loaded into the DAC data register and takes 24 clock cycles.

### 3.6. Signal Quality Dependent on DSP Sampling Frequency

Above, the filter evaluation was carried out at a 10 kHz DSP sampling frequency. In order to assess the influence on the signal quality and power consumption, the DSP sampling frequency was varied in the range from 500 Hz up to 10 kHz. The ADC settings remained unchanged throughout all measurements, and 64 ADC samples were measured per DSP sampling period ($t_{S}=1/f_{S}$). The system clock frequency as well as the DSP sampling frequency were modified according to Section 4.8. The filter parameters were adjusted to the respective DSP sampling frequency. The signal used for the calculation of the SNR was measured with the subjects compressing an adjustable hand grip exerciser allowing an adjustable tension at 20% of the MVC to avoid fatigue from influencing the measurements. Additionally, the subjects took sufficient rest periods and the order of the measurements was inverted for the second subject. The noise level was measured at the relaxed muscle. Two subjects were measured with five sampling frequencies and each measurement was conducted three times. The signal was measured for 5 s with the oscilloscope [42] at a sampling frequency of 20 kHz. The insulated EMG sensor was placed above the flexor carpi radialis and the measurement was started after the onset of the muscle contraction. In the measurements, the implemented notch, highpass and lowpass filters were activated.

## 4. Results

The DSP runtime is significantly reduced by designing the filters particularly for real-time, ultra-low-power applications. No floating-point units and no divisions were implemented, and the number of calculations was minimized for minimal calculation effort. These algorithms were implemented on a low-cost μC.

The final measurement system features IIR filters, as they are more efficient regarding runtime and memory consumption than FIR filters. They can be realized with low filter order, and their delay is negligible. Their parametrization, stability, effect of quantization and runtime were investigated.

### 4.1. Comb Filter

Figure 11 shows the measurement results of the comb filter implementations. For the Butterworth filter with 1 Hz cutoff frequency *f_C_* and for the third-order FIR filter, a peak remains at the PLI frequency. This type of filter has a high filter quality and a steep slope (Figure 6) but does not filter the PLI sufficiently. This is explained by the deviating power-line frequency, which makes a broader comb filter stop-band necessary.

The 5 Hz Butterworth implementation was selected because it provides sufficient PLI filtering and it is insensitive to deviations of the power-line frequency. The Butterworth implementation is preferred over the Chebyshev implementation because it exhibits less overshoot and ringing. The Chebyshev has high gain at the edges of the passband, which has an adverse effect at power-line frequency fluctuations. With the 5 Hz *f_C_* filter, more EMG signal power is preserved than with the 7 Hz *f_C_* filter.

#### Comb Filter Stability

The poles are listed in Table 3 and plotted in Figure 12 to illustrate the filter stability. All filters are stable, as the absolute value is <1 in all implementations. For FIR filters, stability is given in all cases. For IIR filters, poles closer to the origin of the unit circle are preferred in the implementation because oscillations abate more rapidly. The selected 5 Hz Butterworth comb filter fully meets the stability requirements.

### 4.2. Lowpass Filter

In this section, the parametrization of the lowpass filter, which is applied after the highpass filter, is evaluated. For the smoothing lowpass, the architecture, runtime and implementation are equivalent to the filter described in this section, and its parametrization is described in Section 3.4.8.

The lowpass after the highpass filter has the time constant *T* = 300μs and the cutoff frequency $f_{C}$ = 531 Hz. A higher $f_{C}$ does not sufficiently filter the high-frequency edges of the quantizations, and a lower $f_{C}$ would markedly attenuate the EMG signal.

#### Lowpass Filter Stability

Since the pole at 0.75 is located within the unit circle, the filter is stable.

### 4.3. Highpass Filter

Figure 13 shows the highpass-filtered artifacts and contraction EMG. An optimal balance of high artifact attenuation and low EMG signal attenuation is desired. The null hypothesis that the Root Mean Square (RMS) values and the signal loss of the contraction EMG and artifacts are normally distributed was not rejected at a 5% significance level. The one-sample Kolmogorov-Smirnov test [43] resulted in a *p* = 0.9995 for the RMS of the contraction EMG and in a *p* = 0.0896 for the RMS of the artifacts. For the signal loss, the *p*-values depend on the applied filter. The artifacts have lower *p*-values because the probability density function of artifacts is skewed. The standard deviations in Figure 13c result from the frequency characteristics of the highly variable artifacts measured. The signal loss is the difference of the RMS values of the unfiltered and filtered signal, referred to the unfiltered signal. As we strive for low EMG signal loss and high artifact signal loss, the difference of these signal losses is plotted in Figure 13d. Since the maximum is at the cutoff frequency *f_C_* = 60 Hz for the Chebyshev filter, we chose this filter for the final implementation in the insulated EMG measurement system.

#### Highpass Filter Stability

The poles are listed in Table 4 and plotted in Figure 14 to illustrate the stability of the filters. All highpass filters are stable because their poles are located within the unit circle. The poles of the filters with low *f_C_* are close to the edges of the unit circle, but they are stable.

### 4.4. Cascaded Filter Transfer Function

Figure 15 shows the transfer function of the selected comb (5 Hz But), highpass (60 Hz Che) and lowpass ($f_{C}$ = 531 Hz).

### 4.5. Rectification and Smoothing

Linear rectification was chosen for the final implementation, as it enables better proportional control than squared rectification. Since the values of squared high signals are much greater than those of squared small signals, the small signal amplitudes from noise, but also weak contractions, vanish due to the squaring. The slope steepness is better for squaring, but with linear rectification sufficient steepness of the signal slopes still can be achieved for light co-contractions.

The cutoff frequency *f_C_* was set to 3.11 Hz. This lowpass, which is a discrete first-order lag element, has a time constant *T* of 51.1 ms. A lower *f_C_* would lead to longer delays, and a higher *f_C_* would not smooth the signal sufficiently for a stable prosthesis control. Figure 16 shows the rectified and smoothed signals of various movements. As the prosthesis drive has an integrated lowpass characteristic, a slight ripple in the smoothed signal does not affect the drive speed.

### 4.6. Runtime

The runtimes of the implemented signal processing chain for various filters are listed in Table 5. Different lowpass implementations at a 48 MHz clock frequency are compared in Table 6. The runtime is inversely proportional to the clock frequency, i.e., decreasing the clock frequency by half will double the runtime. Avoiding FIR filters is recommended for real-time ultra-low-power systems, as runtime is significantly longer, depending on the order of the filter. Furthermore, the fifth-order FIR filter has a poorer damping characteristic than the presented IIR filter. In many EMG systems, FIR filters have window sizes of 50–350 ms [44], implying a filter order of 500–3500. Such filters cannot be calculated in real-time with this system. The selected implementation requires less calculation effort than the floating-point moving-average filter. Even when the fixed-point implementations are compared, our lowpass architecture decreases runtime.

### 4.7. Power Consumption

The current consumption of the sensor system, as presented by Roland et al. [11] was measured to be 8.5 mA at a supply voltage of 3.7 V (=31.5 mW). This includes the μC, which requires 5.2 mA. For this measurement, the controller was set to 48 MHz clock frequency in PL2 mode and the ADC, DAC, brown out detector and IO-pins were activated. The current consumption of the controller includes also the integrated OpAmps for gain adjustment, the non-volatile memory and the DSP software. The BLE module would increase the current consumption at 3.7 V by 0.9 mA in advertising mode and by 8.9 mA in send/receive mode [45]. The send/receive mode is only activated in case of sensor configuration or data transfer to the PC. In deep sleep mode, the power consumption of the BLE module is negligible. However, the high clock frequency was selected to avoid limitations in the experiments, which comprise also floating point implementations. The relation between μC clock frequency and power consumption is given in Table 7. Due to the fact that the ADC clock is connected to the μC clock, the ADC settings have to be checked and recalculated accordingly.

### 4.8. Signal Quality Dependent on DSP Sampling Frequency

The reduction of the DSP sampling frequency allows a reduction of the clock frequency, which further reduces the power consumption, see Table 7. Figure 17 shows that a reduction of the DSP sampling frequency also reduces the SNR as high frequency noise is reflected to the EMG frequency range.

### 4.9. Memory

The presented implementation including all configurations, filters, rectification and smoothing and the decision algorithm requires 32.4% of the 256 kB program memory and 35.3% of the 32 kB data memory.

### 4.10. Signal Delay

The signal delay is dominated by the time constant *T* of the first-order lag element which forms the lowpass filter for smoothing. This *T* of 51.1 ms is not perceptible to the user as described in the Introduction.

## 5. Discussion

Insulated EMG has various advantages, such as insensitivity to sweat and prevention of pressure marks, over state-of-the-art conductive EMG. However, insulated EMG requires sophisticated measurement electronics and signal processing. Implementing these filters as analog circuits, cost for many components, matching these components and designing and developing a more extensive circuit board arise. Digital filters implemented in a microcontroller based DSP increases flexibility and decreases cost. We suggest digital filtering instead of analog filtering to best meet the requirements for EMG sensing, like low-power consumption, high signal quality, stability, proportionality, etc., as listed in the Introduction.

The comb filter, which suppresses the PLI in this sensor, is very effective, stable and requires low computation power; however, the EMG signal is also reduced. A trade-off between high filter Q and stability has to be dealt with. A high filter Q increases the sensitivity to power-line frequency deviations and, depending on the filter type, causes ringing and overshoot. For stable operation, we suggest applying the 5 Hz Butterworth filter.

One major problem in EMG measurement is movement artifacts. Changes in contact pressure, which occur due to movements, change the coupling capacity and further alter the signal amplitude. Therefore, the evaluated higher cutoff frequency for highpass filtering might also result from the fact that the system is designed for high stability in the context of movement artifacts. We suggest a higher cutoff frequency at 60 Hz for insulated EMG in contrast to the 5–30 Hz cutoff frequency recommended for highpass filtering in conductive EMG.

The EMG signal is smoothed to control the prosthesis drive. A short response time of the system, which is dominated by the *T* (=51.1 ms) of the lowpass filter for smoothing, is desired. A low *T* leads to a fast system while reducing signal quality and stability due to insufficient smoothing. To control the prosthesis drive speed proportional to the muscle force, the prosthesis drive has to be activated already at low signal levels. High slope steepness facilitates co-contractions; however, this impairs stability.

To achieve high signal quality in the evaluations, the DSP sampling frequency was set to 10 kHz. This frequency could be reduced, having a slightly lower signal-to-noise ratio. If the signal is smoothed, high signal quality is not required and we suggest a sampling frequency in the range from 0.5 kHz to 1 kHz. However, if the signal is further processed, like in feature extraction for machine learning, we suggest higher DSP sampling frequencies from 2 kHz up to 10 kHz.

The decision algorithm, which distinguishes between movement artifacts and muscle contraction, has already been tested on amputees in real-world environments. As movement artifacts are a major problem in prosthesis control, this algorithm significantly improves the performance of the prosthesis.

The whole capacitive EMG sensor system has not been applied to amputees in real-world environments yet. Nevertheless, noise, which occurs in real-world environments such as movement artifacts and power-line interferences, were measured and incorporated in the filter design. This work demonstrates the principles of how to implement this filter design aiming at robust control.

## 6. Conclusions

Insulated EMG sensors are different in construction to conductive EMG sensors and can avoid pressure marks and skin irritations. Due to the measurement principle, insulated EMG sensors have special requirements in signal amplification and processing.

DSP implemented into a microcontroller reduces the sensor hardware to a minimum and enables configuration, calibration and continuous improvement without hardware modifications. This work demonstrates optimized system setup, filter selection and implementation into an embedded low-power real-time sensor. By incorporating artifacts and other noise, which occur in real-world environments, to the selection of the filter parameters, a robust system was obtained. High signal quality and stable control of the myoelectric upper-limb prosthesis were achieved with these algorithms, although they require minimal calculation effort.

With the knowledge of the DSP calculation effort, the clock frequency and the power consumption can be optimized. This microcontroller based sensor system provides the opportunity to implement and test future innovations and developments.

As part of future work, we plan to further investigate the discrimination between artifacts and muscle contraction to ensure that the prosthesis drive is activated only by actual muscle contractions. Time domain features allow fast decisions at minimal calculation effort. Our next step will be the testing of the presented insulated EMG measurement system with amputees in real-world environments.

To extend the system to a multi-sensor application, we will design low-power decision algorithms to distinguish between various hand movements for high-dexterity prostheses.

## Figures and Tables

**Figure 1 sensors-19-00959-f001:**
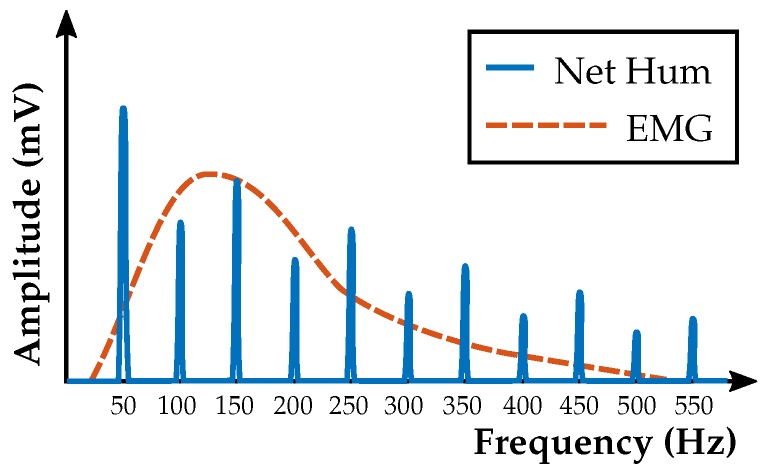
Qualitative sketch of the power-line frequency and its harmonics.

**Figure 2 sensors-19-00959-f002:**
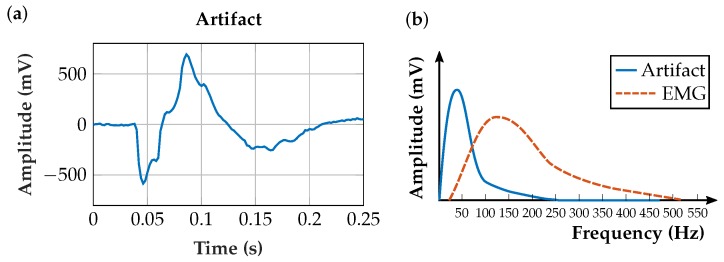
(**a**) A typical measured movement artifact in the time domain; (**b**) qualitative sketch of a typical movement artifact located in a lower frequency range than the contraction EMG.

**Figure 3 sensors-19-00959-f003:**
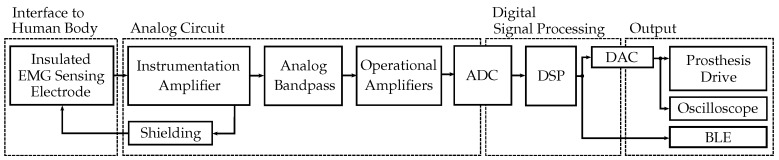
Block diagram outlining the insulated EMG measurement set-up. First, the EMG signal coupled from the human body is amplified and filtered by the analog circuit. In the μC, the signal is then digitally filtered and provided at the sensor system output.

**Figure 4 sensors-19-00959-f004:**
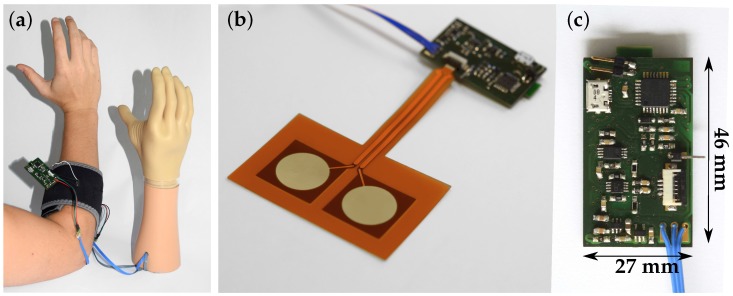
(**a**) Set-up of the capacitive EMG measurement system to control a myoelectric upper-limb prosthesis. The EMG sensor is fixed to the human forearm by a cuff; (**b**) the flexible sensor is attached to the circuit board; (**c**) front side of the circuit board.

**Figure 5 sensors-19-00959-f005:**
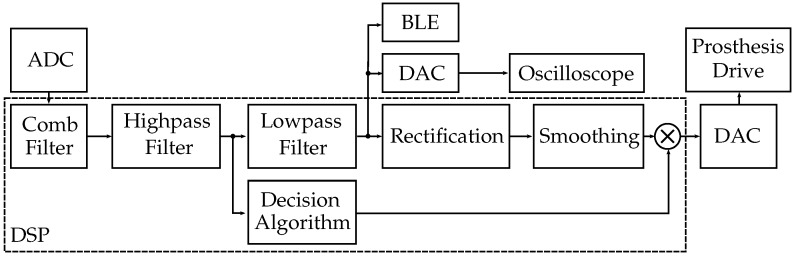
Block diagram of the DSP path, 50 Hz comb filter followed by a highpass, a lowpass and a decision algorithm that handles the movement artifacts. Rectification and smoothing are required for controlling the prosthesis drive. For experiments, the signal can also be connected to an oscilloscope via the digital-to-analog converter (DAC) or transmitted via the Bluetooth Low Energy (BLE) module.

**Figure 6 sensors-19-00959-f006:**
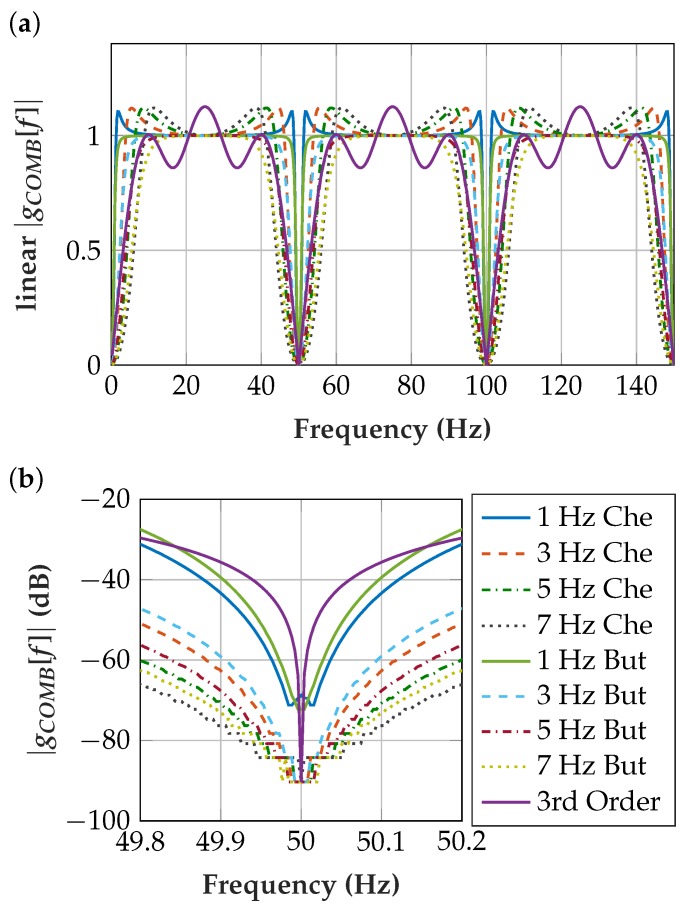
(**a**) Comb filter transfer function showing the passband behavior (linear *y*-axis); (**b**) magnification of comb filter transfer function with quantization effects showing the stop-band behavior (*y*-axis in dB).

**Figure 7 sensors-19-00959-f007:**
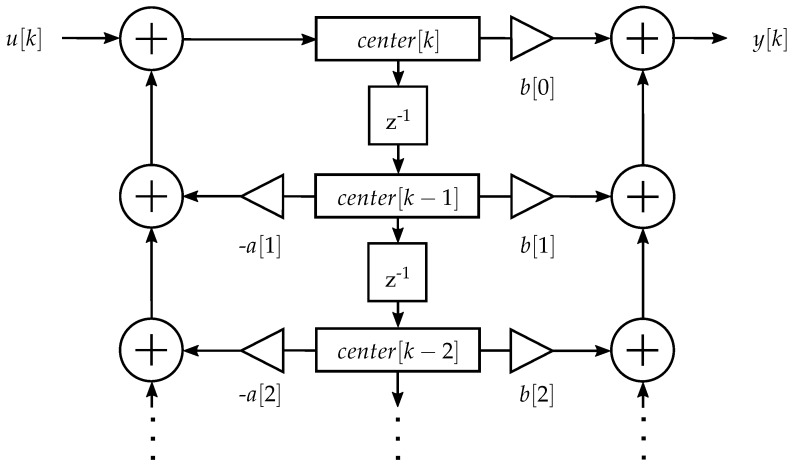
Signal flow graph of a second-order IIR direct form II filter realization, with the numerators *b* and the denominators *a*. Only one storage array, center, is required.

**Figure 8 sensors-19-00959-f008:**
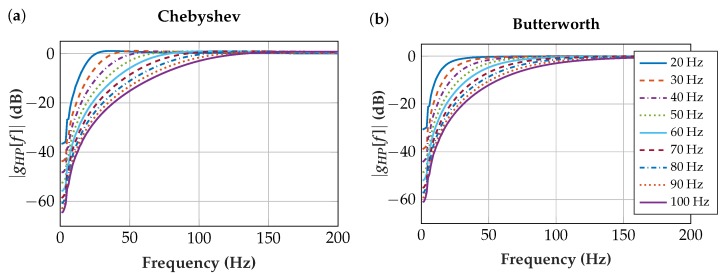
Transfer functions with quantization effects included. (**a**) Chebyshev highpass filter; (**b**) Butterworth highpass filter.

**Figure 9 sensors-19-00959-f009:**
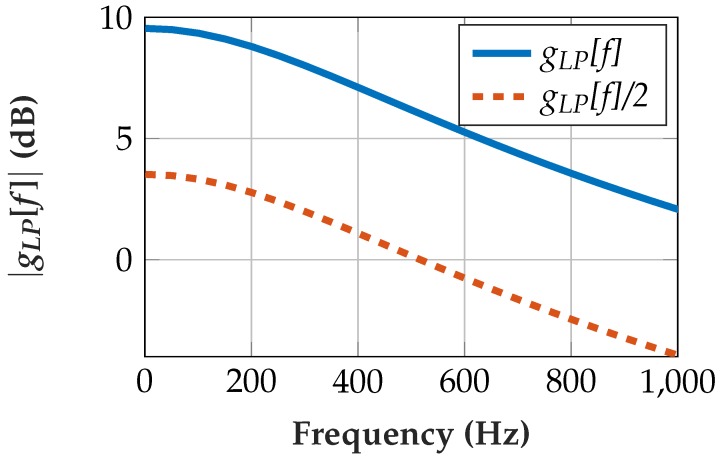
Transfer function |*g_LP_*[*f*]| of lowpass with $f_{C}$ = 531 Hz. One right-shift (division by 2) for scaling of the preliminary output to prevent overflows in the implementation.

**Figure 10 sensors-19-00959-f010:**
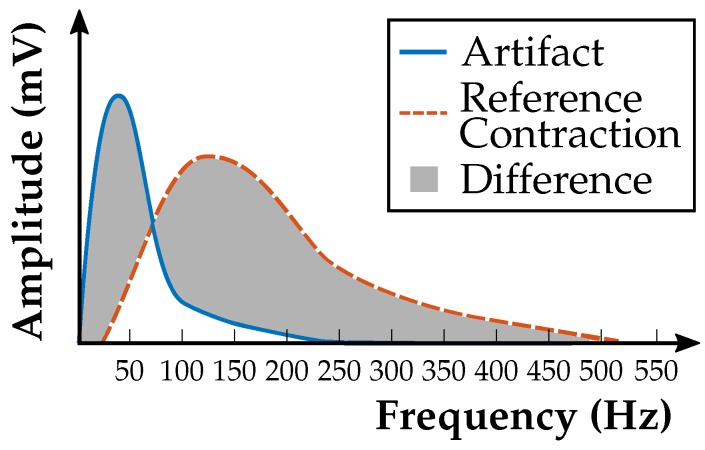
Principle of calculating the difference between current and reference STFT to distinguish between artifact and contraction signal. The prosthesis drive is activated or disabled based on this decision [23].

**Figure 11 sensors-19-00959-f011:**
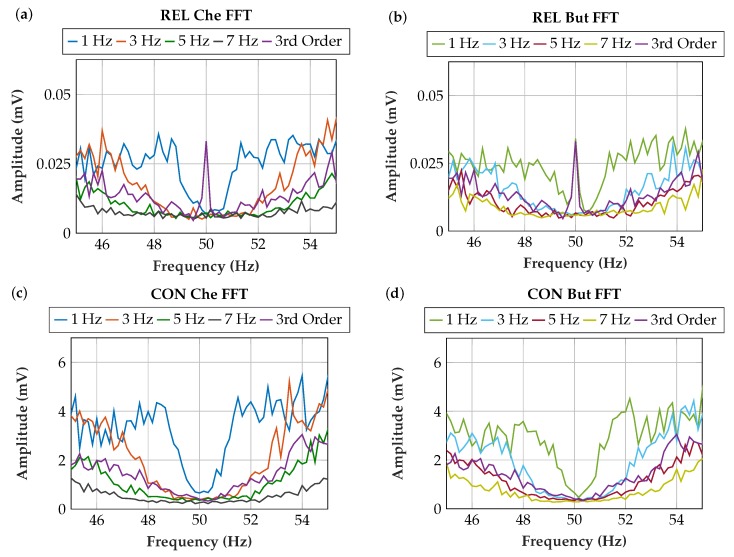
Measurement results of various comb filter implementations (magnification in frequency domain); (**a**) Chebyshev-filtered signal at relaxed muscle; (**b**) Butterworth-filtered signal at relaxed muscle; (**c**) Chebyshev-filtered signal at contracted muscle; (**d**) Butterworth-filtered signal at contracted muscle.

**Figure 12 sensors-19-00959-f012:**
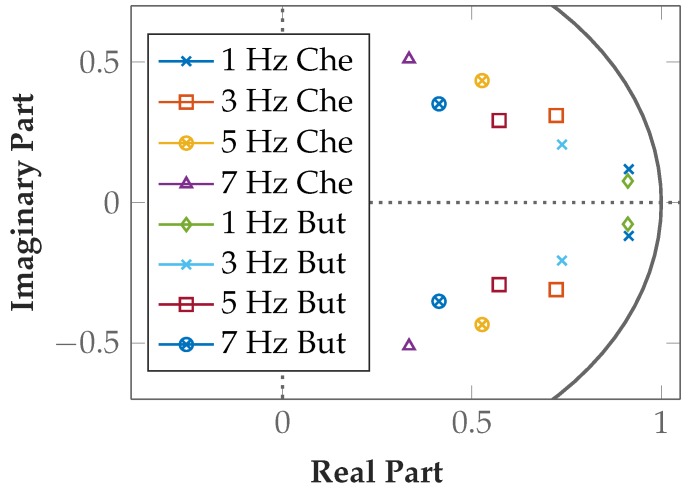
Poles of Chebyshev and Butterworth comb filter implementations are located within the unit circle.

**Figure 13 sensors-19-00959-f013:**
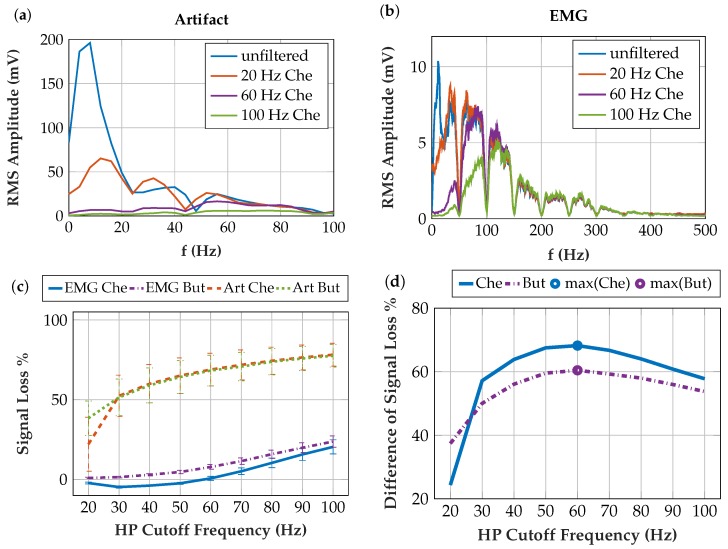
(**a**) unfiltered and filtered artifacts in the frequency domain (Chebyshev filters with different *f_C_*); (**b**) unfiltered and filtered contraction EMG in the frequency domain; (**c**) signal loss in % when filtering artifact and contraction EMG. The signal loss in % can be negative due to the positive gain at some frequencies at the Chebyshev filter; (**d**) difference between artifact and EMG signal losses with an optimum at the 60 Hz Chebyshev filter.

**Figure 14 sensors-19-00959-f014:**
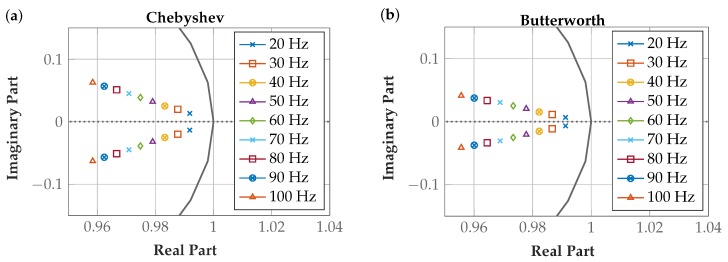
Poles of highpass filter implementations are located within the unit circle. (**a**) Chebyshev filters; (**b**) Butterworth filters

**Figure 15 sensors-19-00959-f015:**
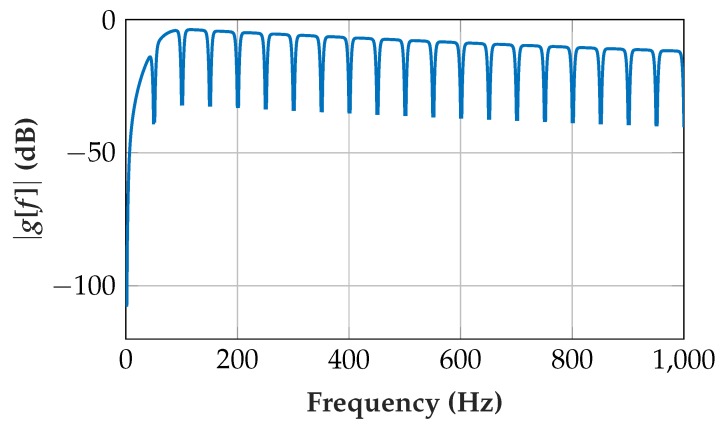
Transfer function of the cascaded comb, highpass and lowpass filter (including compensation of scaling *S*).

**Figure 16 sensors-19-00959-f016:**
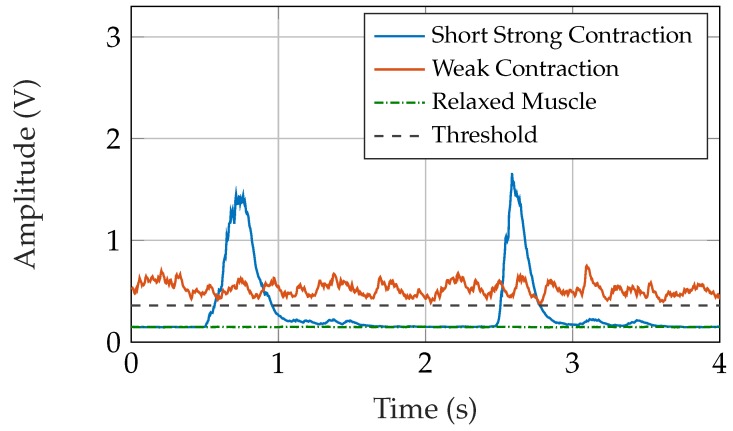
Rectified and smoothed EMG signal for various movements with the threshold for prosthesis drive activation indicated.

**Figure 17 sensors-19-00959-f017:**
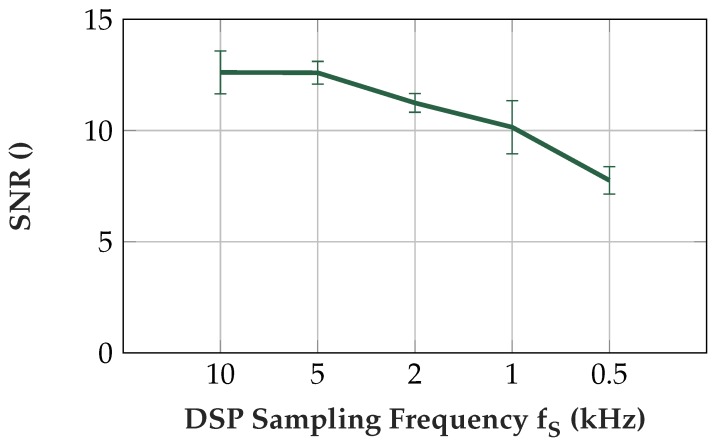
SNR at different sampling frequencies. The EMG signal was measured at a muscle force at 20% of MVC and the noise was measured at a relaxed muscle. Note that the SNR is higher at 100% MVC than at these measurements with lower muscle force.

**Table 1 sensors-19-00959-t001:** Floating-point comb filter coefficients.

Filter	a[0]	a[200]	a[400]	a[600]	b[0]	b[200]	b[400]	b[600]
*1 Hz Che*	1	−1.8278	0.8501	-	1	−2	1	-
*3 Hz Che*	1	−1.4449	0.6186	-	1	−2	1	-
*5 Hz Che*	1	−1.0531	0.4665	-	1	−2	1	-
*7 Hz Che*	1	−0.6680	0.3730	-	1	−2	1	-
*1 Hz But*	1	−1.8227	0.8372	-	1	−2	1	-
*3 Hz But*	1	−1.4755	0.5869	-	1	−2	1	-
*5 Hz But*	1	−1.1430	0.4128	-	1	−2	1	-
*7 Hz But*	1	−0.8252	0.2946	-	1	−2	1	-
*3rd order*	1	0	0	0	1	−0.5	−0.25	−0.25

**Table 2 sensors-19-00959-t002:** Floating-point highpass filter coefficients.

Filter	a[0]	a[1]	a[2]	b[0]	b[1]	b[2]
*20 Hz Che*	1	−1.9836	0.9839	1	−2	1
*30 Hz Che*	1	−1.9754	0.9759	1	−2	1
*40 Hz Che*	1	−1.9664	0.9674	1	−2	1
*50 Hz Che*	1	−1.9580	0.9596	1	−2	1
*60 Hz Che*	1	−1.9496	0.9518	1	−2	1
*70 Hz Che*	1	−1.9418	0.9447	1	−2	1
*80 Hz Che*	1	−1.9333	0.9370	1	−2	1
*90 Hz Che*	1	−1.9247	0.9294	1	−2	1
*100 Hz Che*	1	−1.9168	0.9225	1	−2	1
*20 Hz But*	1	−1.9824	0.9825	1	−2	1
*30 Hz But*	1	−1.9733	0.9737	1	−2	1
*40 Hz But*	1	−1.9645	0.9651	1	−2	1
*50 Hz But*	1	−1.9556	0.9565	1	−2	1
*60 Hz But*	1	−1.9467	0.9481	1	−2	1
*70 Hz But*	1	−1.9378	0.9397	1	−2	1
*80 Hz But*	1	−1.9289	0.9314	1	−2	1
*90 Hz But*	1	−1.9201	0.9231	1	−2	1
*100 Hz But*	1	−1.9112	0.9150	1	−2	1

**Table 3 sensors-19-00959-t003:** Poles of comb filters with quantized coefficients.

$f_{C}$	1 Hz	3 Hz	5 Hz	7 Hz
Chebyshev	0.914 ± 0.119i	0.723 ± 0.310i	0.527 ± 0.435i	0.335 ± 0.511i
Butterworth	0.912 ± 0.077i	0.738 ± 0.206i	0.572 ± 0.292i	0.413 ± 0.352i

**Table 4 sensors-19-00959-t004:** Poles of highpass filters with quantized coefficients.

$f_{C}$	**20 Hz**	**30 Hz**	**40 Hz**	**50 Hz**	**60 Hz**
Che	0.992 ± 0.013i	0.988 ± 0.020i	0.984 ± 0.025i	0.980 ± 0.032i	0.976 ± 0.039i
But	0.991 ± 0.007i	0.987 ± 0.011i	0.982 ± 0.015i	0.978 ± 0.021i	0.974 ± 0.025i
$f_{C}$	**70 Hz**	**80 Hz**	**90 Hz**	**100 Hz**	
Che	0.972 ± 0.045i	0.968 ± 0.051i	0.964 ± 0.057i	0.961 ± 0.057i	
But	0.969 ± 0.030i	0.965 ± 0.033i	0.960 ± 0.037i	0.956 ± 0.041i	

**Table 5 sensors-19-00959-t005:** Runtime per sample of filters in C implementation at a 48 MHz clock.

	(μs)
Comb Filter (2nd order IIR)	8.17
Highpass Filter (2nd order IIR)	5.88
Lowpass Filter (1st order IIR)	1.28
Rectification and Smoothing	2.62
Total Signal Processing Chain Runtime	17.95

**Table 6 sensors-19-00959-t006:** Comparison of runtime per sample of various lowpass-filter C implementations at a 48 MHz clock.

	(μs)
(i) Selected Lowpass Filter (1st order IIR)	1.28
(ii) Fixed-point Lowpass Filter (1st order IIR, direct form II)	1.67
(iii) Floating-Point Lowpass Filter (1st order IIR, direct form II)	20.12
(iv) Floating-Point Lowpass Filter (5th order FIR)	49.80
(v) Floating-Point Lowpass Filter (8th order FIR)	73.60

**Table 7 sensors-19-00959-t007:** Effect of reducing sampling and clock frequency on power consumption and signal quality.

Sampling Frequency $f_{S}$ (kHz)	μC Clock Frequency (MHz)	Power Consumption (mW)	SNR
10.0	48	31.5	12.6
5.0	16	20.5	12.6
2.0	8	15.9	11.2
1.0	4	14.4	10.1
0.5	2	13.7	7.8

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
