# Peer review of "Ultra-Low-Power Digital Filtering for Insulated EMG Sensing"

_sensors, 2019, doi:10.3390/s19040959_

Round 1

Reviewer 1 Report

The paper explain the designing of digital filters to use in EMG signals.

I suggest to the authors to include any picture of the real system and not only block diagrams, this pictures could helps the readers to undertans the real use of their desing.

I also suggest the authors to check the paper to avoid the repeated parts about the necesary low power use, the fixed points system... after reading the "introduction" and the "problem definition" I think that I read twice (at least) the good thinks of their design.

Moreover, I miss the explanation about why they use digital filters. Figure 3 shows an Analog Bandpass filter, and I don't understand why the other filters could not be analog too; it could improve the speed of the uC. Also I suggest the author to check the possibility of reducing the clokc frecuency; the said that could be reduce to decrease the power consumption, but in most of the uC, this reduction affects the sample frecuency of the ADC.

I think that the explanation of the filter design is very good, but when I read the abstract I think that the paper would talk more about the EMG and its use, and not as much about filter design.

Author Response

Response to Reviewer 1 Comments

Thank you for your appreciation of our work and thank you for your comments, which helped us with improving our work. Please find below a point-by-point response to your comments.

Point 1: The paper explain the designing of digital filters to use in EMG signals. I suggest to the authors to include any picture of the real system and not only block diagrams, this pictures could helps the readers to undertans the real use of their desing.

Response 1: In the revised manuscript pictures of the real system are included.

Point 2: I also suggest the authors to check the paper to avoid the repeated parts about the necesary low power use, the fixed points system... after reading the "introduction" and the "problem definition" I think that I read twice (at least) the good thinks of their design.

Response 2: You are right, there were a lot of repetitions, thank you for that comment. These repetitions were deleted, as suggested.

Point 3: Moreover, I miss the explanation about why they use digital filters. Figure 3 shows an Analog Bandpass filter, and I don't understand why the other filters could not be analog too; it could improve the speed of the uC. Also I suggest the author to check the possibility of reducing the clokc frecuency; the said that could be reduce to decrease the power consumption, but in most of the uC, this reduction affects the sample frecuency of the ADC.

Response 3: The advantages of using digital over analog filters are included now in the introduction. The effect of reducing clock frequency on power consumption and sampling frequency is evaluated, as suggested.

Point 4: I think that the explanation of the filter design is very good, but when I read the abstract I think that the paper would talk more about the EMG and its use, and not as much about filter design.

Response 4: In the revised manuscript, parts of the abstract are re-written to avoid misleading of the reader.

Reviewer 2 Report

Authors developed ultra-low-power digital signal processing algorithms for EMG sensor. What presented is a good try, but as a scientific paper, the authors should pay much more efforts, to highlight the novelty, independent validation, to have a focus, to be more scientific, to be more specific and to discuss the limitation. I would say this is not a well written manuscript with many errors of execution or presentation. This reviewer carefully read this manuscript and found some critical and major concerns those are raised below:

1. First of all, the manuscript needs tremendous amounts of help regarding grammar and writing style. It was very hard to understand most of the concepts due to this fact and made reviewing this paper challenging at the least. Please check with native English writer. In summary, the writing is not always well structured, and it is hard for the reader to get the take home message.

2. The introduction (rationale) and discussion (interpretation) need to be strengthen to highlight the significance of the present investigation. In fact, there is no Discussion section in the manuscript.

3.  The rationale needs to be more convincing.  As it stands, the main thrust of the introduction is that previous research in this area is limited, which need to highlight more for convincing reason for why it should be conducted now.  I suggest you clearly explain the benefit to the practitioner and show how this study can be developed and incorporated to inform, evaluate, and design training. For example, one important feature is “low-cost”, but there is no information about the cost of existing system(s).

4. The abstract is very weak. Researches will read the abstract to decide whether your article pertains to their interests and needs.

5. Author claimed the developed device can be used in real-world environment, therefore my question, have you tested in the real-world (free-living) which means outside of the laboratory?

6. What was the justification to choose forearm muscle instead of most widely and renowned EMG measuring muscle Biceps brachii?

7. Page 5, Line 145-156: “The EMG sampling rate 10 kHz…”, is this sampling rate too high for EMG?

8. Need a good explanation about validation of the system.

9. Were the data normally distributed? Was there a check for this?

10. Of the references used, several rather weak papers are used to support important statements

Author Response

Response to Reviewer 2 Comments

Thank you for your appreciation of our work and thank you for your comments, which helped us with improving our work. Please find below a point-by-point response to your comments.

Point 1: First of all, the manuscript needs tremendous amounts of help regarding grammar and writing style. It was very hard to understand most of the concepts due to this fact and made reviewing this paper challenging at the least. Please check with native English writer. In summary, the writing is not always well structured, and it is hard for the reader to get the take home message.

Response 1:  The paper has undergone scientific language editing. A native English speaker read and corrected the manuscript and is now added to the acknowledgments. The take home message was highlighted to ensure that the reader gets the concepts presented in the paper.

Point 2: The introduction (rationale) and discussion (interpretation) need to be strengthen to highlight the significance of the present investigation. In fact, there is no Discussion section in the manuscript.

Response 2: Paragraphs, which strengthen the significance of the present investigation, are added to the introduction. A discussion section is added to the revised version of the manuscript.

Point 3: The rationale needs to be more convincing.  As it stands, the main thrust of the introduction is that previous research in this area is limited, which need to highlight more for convincing reason for why it should be conducted now.  I suggest you clearly explain the benefit to the practitioner and show how this study can be developed and incorporated to inform, evaluate, and design training. For example, one important feature is “low-cost”, but there is no information about the cost of existing system(s).

Response 3: Reasons for why it should be conducted now are added to the introduction. These reasons point out the benefits for the practitioner when using the proposed system. Further quantitative results are added to substantiate the benefits of the system. Information about cost is now included to the paper. Differences in cost were not given, as it would be misleading to compare a prototype with a commercial EMG sensor that had to undergo medical certification.

Point 4: The abstract is very weak. Researches will read the abstract to decide whether your article pertains to their interests and needs.

Response 4: The abstract is re-written and the benefits of the system are strengthened. In the revised manuscript, the abstract represents the paper in a better way now.

Point 5: Author claimed the developed device can be used in real-world environment, therefore my question, have you tested in the real-world (free-living) which means outside of the laboratory?

Response 5: The decision algorithm has already been tested in real-world environment. As movement artifacts are a major problem in biosignal acquisition, it significantly improves performance in prosthesis control.

Movement artifacts are considered in the evaluation of the optimal filter parameters to obtain a robust system for real-world environments. These artifacts were created by mechanical interferences, which might occur in real-world environments, such as tapping, shifting or lifting of the sensor. By using these artifacts for evaluating the filter parameters, the resulting system will be robust against these interferences.

The whole capacitive EMG sensor system was not applied to amputees in real-world environments yet. Nevertheless, interferences, which occur in real-world environment such as movement artifacts, power-line interferences and noise were measured and incorporated in the filter design. This work demonstrates the principles of how to implement the filter design aiming at robust control.

This information is added to the revised manuscript.

Point 6: What was the justification to choose forearm muscle instead of most widely and renowned EMG measuring muscle Biceps brachii?

Response 6: The forearm muscle was used, as a large proportion of upper-limb amputations are at the trans-radial level or farther distal. This information, including references, is added to the paper. The operation principle of capacitive measurement at biceps brachii is the same.

Point 7: Page 5, Line 145-156: “The EMG sampling rate 10 kHz…”, is this sampling rate too high for EMG?

Response 7: Indeed, a sampling frequency of 10 kHz is high for EMG. However, for high signal quality, it is suggested to use high sampling frequencies. For evaluation of the optimal filter parameters, high signal quality is desired. The effects of reducing the sampling rate on signal quality, clock frequency and further power consumption is investigated and added to the revised manuscript.

Point 8: Need a good explanation about validation of the system.

Response 8: In the revised manuscript, further quantitative findings are added to the results section to strengthen the validation of the system.

Point 9: Were the data normally distributed? Was there a check for this?

Response 9: The data are normally distributed as tested with a Kolmogorov-Smirnov test. This information is added to Section 3.3.

Point 10: Of the references used, several rather weak papers are used to support important statements.

Response 10: References were added to support important statements, as suggested.

Besides the references in the color boxes, following topics were supplemented with stronger references:

state-of-the-art electrodes,

preference of a robust over a complex, yet unreliable prosthesis,

commercially available systems (Otto Bock product catalogue),

EMG frequency range,

movement artifacts in low frequency range (< 20 Hz).

Round 2

Reviewer 1 Report

I think that the article improves considerately after the changes made by the authors, and now it's easier to understanding.

Reviewer 2 Report

The authors carefully revised the manuscript and addressed all issues raised with respect to the previous version in a satisfactory way. The updated manuscript is now satisfactory in all aspects.